# A Deep-Learning-Based Guidewire Compliant Control Method for the Endovascular Surgery Robot

**DOI:** 10.3390/mi13122237

**Published:** 2022-12-16

**Authors:** Chuqiao Lyu, Shuxiang Guo, Wei Zhou, Yonggan Yan, Chenguang Yang, Yue Wang, Fanxu Meng

**Affiliations:** 1School of Life Science, Beijing Institute of Technology, Beijing 100081, China; 2Department of Mechanical Engineering, Tsinghua University, Beijing 100084, China

**Keywords:** endovascular surgery robot, deep learning, compliant control, force feedback, guidewire

## Abstract

Endovascular surgery is a high-risk operation with limited vision and intractable guidewires. At present, endovascular surgery robot (ESR) systems based on force feedback liberates surgeons’ operation skills, but it lacks the ability to combine force perception with vision. In this study, a deep learning-based guidewire-compliant control method (GCCM) is proposed, which guides the robot to avoid surgical risks and improve the efficiency of guidewire operation. First, a deep learning-based model called GCCM-net is built to identify whether the guidewire tip collides with the vascular wall in real time. The experimental results in a vascular phantom show that the best accuracy of GCCM-net is 94.86 ± 0.31%. Second, a real-time operational risk classification method named GCCM-strategy is proposed. When the surgical risks occur, the GCCM-strategy uses the result of GCCM-net as damping and decreases the robot’s running speed through virtual resistance. Compared with force sensors, the robot with GCCM-strategy can alleviate the problem of force position asynchrony caused by the long and soft guidewires in real-time. Experiments run by five guidewire operators show that the GCCM-strategy can reduce the average operating force by 44.0% and shorten the average operating time by 24.6%; therefore the combination of vision and force based on deep learning plays a positive role in improving the operation efficiency in ESR.

## 1. Introduction

Cardiovascular and cerebrovascular diseases are serious health threats. Endovascular surgery (ES) has been widely used as an effective treatment. Compared with open vascular surgery, patients treated with ES have less trauma and faster postoperative recovery. However, because of high radiation, ES is not favored by surgeons [1]. Therefore, the study of an endovascular surgery robot (ESR) system has great significance [2]. An ESR system consists of two parts: master end and slave end. The master end collects the actions of the surgeon, and the slave end repeats these actions. Robots can provide teleoperation force feedback and surgical images for surgeons, thereby reducing the radiation exposure. In recent years, a series of commercialized ESR systems have achieved good clinical response, such as Corpath GRX [3] and Sensei X [4].

As the endovascular environment is narrow and fragile, slender guidewires must be used for the operation. Some studies [5] showed that the flexible guidewires have better ability to pass through narrow positions, while rigid guidewires have better mobility in open space. To ensure the skillful control of guidewires, certain studies have suggested that the haptic feedback is a valuable research direction [6,7,8,9,10]. For instance, Shi et al. [6] proposed an ESR system with a spring-based haptic interface. In their research, the collision detection algorithm was used to make the guidewire move in a safe force range. Bao et al. [7,9] designed a strategy of multilevel concepts for operating force. In in vitro and in vivo experiments, safe operation was realized using the robot-assisted force control. Jin et al. [8] established an elastic model of vascular walls. A pressure-sensitive rubber sensor was used at the catheter tip to effectively detect the pressure. Other micro sensors include strain gauges [11] and optical fibers [12]. The advantage is that they can obtain tactile force in time, but Hooshiar et al. [10] indicated that the size and electrophysiological adaptability of these sensors need careful consideration.

Schneider et al. [13] suggested that the limited visual field is not conducive to the popularization of surgical methods. Therefore, in recent years, many vision-based methods are deployed on the robot [14,15,16,17]. For instance, Guan et al. [14] used deep learning and transfer learning techniques to match computed tomography angiography (CTA) images with digital subtraction angiography (DSA) images, and realized 2D-3D cerebrovascular reconstruction. Dagnino et al. [15] proposed that tracking the dynamic visual movement of the interaction between the vascular wall and the catheter tip can minimize potential damage. Razban et al. [16] used the vision feedback of guidewire-vascular interaction and numerical finite element modeling (FEM) to calculate the stress. Li et al. [17] proposed a method which combines the YOLOv3 and SA-hourglass algorithm to localize the tip of moving guidewire. Their method also located the position of other surgical instruments in real-time fluorescences. However, few studies have considered both surgical vision and surgical strength, but experienced surgeons can have both. There have been many innovative studies about automated ESR systems. For instance, Chi et al. [18,19] used reinforcement learning to optimize the catheter insertion trajectory. There are many steerable guidewires researches mentioned [20]. In this study, only the passive guidewire is considered.

Thus, the research purpose of ESR system is to improve surgeons’ perception of the state of surgery, and then improve the safety of operations. One of the challenges is to ensure the safety of the operation given the limited vision. The mainstream method is to use the fluorescence imaging to assist endovascular surgery in real time. During this procedure, the patient’s blood vessels are always invisible, because their density is low. In contrast, the material density of the guidewire is high, and its movement and abnormal bending state can be clearly seen, as shown in Figure 1b. Clinically, the way to observe the outline of blood vessel is to inject contrast agents. However, due to the effect of blood flow and kidney, the duration of the vascular artifact is very short, only a few tens of seconds. Even though the vessel walls can be captured by the DSA method, it is not in real-time, and cannot be embedded in the robot-based protection methods. At the same time, excessive injection of contrast agents can cause kidney damage [21]. In most cases, surgeons rely on knowledge of anatomy and soft-tipped guidewires to attempt access to vessel branches. This will inevitably lead to a “hidden collision” behavior, as shown in Figure 1a.

Here, the “hidden collision” occurs when the guidewire tip comes in contact with the vascular wall and ends when it is completely withdrawn and restored to its original shape. Intuitively, this characteristic is influenced by manipulation, vessel contour, and guidewire stiffness. It is hard to be identified by a computational method but can be easily identified by human eyes. Some studies [16] have calculated operating forces based on the degree of guidewires’ abnormal bending (i.e., deflection), but this approach cannot accommodate complex vasculature. In this aspect, methods based on deep learning are more extensible. Experienced surgeons can detect the “hidden collision” in time, improve the control precision of the guidewire, and reduce the risk of surgery. However, long-term concentration may lead to fatigue, so making a real-time recognition model is reasonable.

To sum up, this paper established a deep-learning-based guidewire compliance control model (GCCM-net), which used to discriminate the “hidden collision” behavior of the guidewire tip in an invisible vascular environment. The real-time recognition accuracy of GCCM-net reached 94.86%, and the inference speed was about 89.72 Hz (with CUDA acceleration). Furthermore, we used the results to classify the risks of real-time surgery. A GCCM-strategy was designed in which the haptic robot will influence the operator’s behavior when a “hidden collision” occurs. The results showed that compared with the sensor-based force feedback method, the strategy-based force feedback behavior reduced the average operating force by 44%, and shortened the average operating time by 24.6%.

The ESR experimental platform with invisible vascular environment is conceived and developed in Section 2. The GCCM-net is built in Section 3. The GCCM-strategy is described and evaluated in Section 4. In Section 5, the applications and limitations of the proposed GCCM are systematically discussed. The conclusions are presented in Section 6.

## 2. Experimental Platform

The methodology proposed in this study is based on an ESR experimental platform and shown in Figure 2. The robot platform consists of three parts, which are a slave end, a master end, and a vascular phantom.

### 2.1. Slave End

For the slave end, some components are redesigned and fabricated based on the proposed method [22,23]. The module structure is shown in Figure 3. The grasper in Figure 3b is extremely small and can maintain the guidewire during movement. A load cell below the grasper can capture the haptic force when the grasper tends to move. The guidewire rotation is driven by a servomotor (EC-max 16 and a driver of EPOS2 50/5, manufactured by Maxon Co., Ltd., Tokyo, Japan), as shown in Figure 3a. The torsion force is not considered in this study. The whole module is approximately 10 cm long and deployed on a ball screw, which is driven by second servomotor (SKR/KR26, with the travel distance of 300 mm, manufactured by THK Co., Ltd., Tokyo, Japan) shown in Figure 3c,d. Therefore, the functional movement space of the robot motion is approximately 20 cm. This range can guarantee the free movement of the guidewire in the proposed vascular phantom.

### 2.2. Master End

At the master end, the robot states are controlled by triggering different keys that can, among other things: (1) adjust whether the blood vessels are visible or not; (2) switch the sensor-based force feedback mode to the strategy-based force feedback mode. These two functions are used later as the control variable conditions of the experiment. A commercial haptic device (Geomagic TouchX) is used to capture hand movements and provide haptic feedback. It has six degrees of freedom (DoF) motions and three DoF feedback forces, which is a customary feature among surgical robots. The whole master end is shown in Figure 4a. Where $M_{x}$ and $M_{y}$ is obtained from the Geomagic TouchX’s handle, and they represent the translation and rotation operations of the operator’s hand, respectively. The master-slave adopts the controller area network (CAN) bus and uses the interpolated position method to control the motors. At the slave end, the rotation and translation of the robot are denoted by $S_{a}$ and $S_{b}$, respectively, as shown in Figure 3d. The master end consists of a monitor, whose functions include surgical motion monitoring, motion constraints, and force visualization, as shown in Figure 4b. Surgical monitoring refers to capturing the operation video through the camera (4K resolution, manufactured by S-YUE Co., Ltd., Beijing, China) above the vascular model and then showing the guidewire in the image. This method aims to simulate the natural environment. For motion constraints, we added force feedback at both ends of the maximum motion space to ensure the safety of the robot. For force visualization, the feedback forces are displayed as real-time dynamic curves. The purpose of visualization is to remind the operator to observe the surgical state and respond in time.

### 2.3. Vascular Phantom

The vascular phantom in this study is made of transparent acrylic, as shown in Figure 5. The vascular contour was modeled after a real patient’s aortic arch. In the vascular phantom, the widest area is approximately 3 cm, and the narrowest is less than 1 cm, the thickness of the vessel wall is approximately 1 cm, and the guidewire can rotate freely in it. Considering that the vascular wall is elastic and easy to deform, in this study, soft white clay was used to cover specific positions, as the white triangles shown in Figure 5. This has two functions: (1) Increase the deformability of the vessel and ensure that the deformation of guidewires at the same position is different. (2) Change the contour of the vascular wall and simulate the specificity of vascular branches and lesions. The specificity of the vascular phantom was improved using the above method. The “hidden collisions” occur frequently. In the next section, we provide an accurate definition of "hidden collisions" samples and establish a deep learning model to identify them.

## 3. Hidden Collisions Detection

This section concerns the deformation state of the moving guidewire. For surgeons, the “hidden collisions” cannot avoid owing to the error in judgements caused by the fatigue of long-term operation. For the ESR system, a real-time “hidden collisions” detection model is required. Therefore, in this section, a model named GCCM-net is proposed. We compared the GCCM-net with a non-deep learning model and the state-of-the-art model. The “Ablation Studies” and “Class Activation Mapping” were also used to evaluate the proposed methods.

### 3.1. Datasets

In real surgery, the guidewires always must be pre-shaped, so the guidewire tip is not always in a position where “hidden collisions” occur. Limited by the experimental conditions, a standard “J-type” guidewire is selected. Here we assume that “hidden collisions” always occur at the guidewire tip. Therefore, a single image can be clearly marked as “1” or “0” according to whether the guidewire tip is in contact with the vascular wall or not. So here the “hidden collisions” represent a continuous state, that is, from the moment when the guidewire tip touches the vascular wall to the moment when it retracts.

An operator with no medical experience was invited and allowed to control the ESR system. The operation goal was to determine and cross the soft white clays in an invisible vascular environment. If the operator wants to achieve the target, the “hidden collisions” cannot be avoided. After 60 min of operating, We collected a dataset containing 72,331 images (20 Hz), named $D_{frame}=\left\{f_{1},f_{2},f_{3},\cdots ,f_{72331}\right\}$. Here each $f_{i}$ represented a frame captured when the guidewire was operating in the blood vessel. Next, frames in which the guidewire tip touches the vessel were marked as “1” and the frames with no contact were marked as “0”. A labeled set $P_{frame}=\left\{P_{1},P_{2},P_{3},\cdots ,P_{72331}\right\}$ was also created. In $P_{frame}$, the ratio of the positive samples $\left(P_{frame}=0\right)$ to the negative samples $\left(P_{frame}=1\right)$ was approximately 65.95%. The threshold algorithm was used to extract the guidewire contour and resize the image to 256×256, as shown in Figure 5c. When the vascular environment is invisible, a single image cannot directly show the “hidden collisions” behavior. Therefore, in this study, consecutive 2D frames are regarded as one 3D sample. Only when all frames in $P_{frame}$ are equal to “1”, the 3D sample will be regarded as a “hidden collisions” sample. Otherwise, it still a non-collision sample. As shown in Figure 5d,e, the images {a,b,c,d} are non-collision samples, and the images {e,f,g,h} are “hidden collisions” samples. Note that for visibility, we stacked the images of 3D samples on one image.

Evidently, the “hidden collisions” sample is more prone to irregular deformation than the non-collision sample. Because when the guidewire tip moves in a non-collision sample, most of the guidewire profiles coincide. However, in “hidden collision” samples, the middle section of the guidewire is easily bent because the tip is restrained, which also leads to the generation of shadows. These distinguishing features are unstructured, and suitable to be used to establish a binary classification task based on deep learning, just like the MNIST dataset [24].

Some studies [25] show that the duration, sampling interval, and frame interval affect the real-time classification results. Therefore, the $D_{frame}$ is re-divided according to sampling interval and sampling duration, and the new datasets $D_{sp1}$, $D_{sp2}$, $D_{sp3}$ and $D_{total}$ (where sp means spatiotemporal, and total means the combination of three datasets) are shown in Table 1.

Here $T_{f}$ refers to frame interval, $T_{s}$ refers to sampling duration, and $T_{d}$ refers to sampling interval. According to the camera’s resolution of 20 Hz, 1f is approximately 0.05 s. Every $P_{sp}$ corresponds to four consecutive frames. If all labels in $P_{frame}$ are zero, $P_{sp}$ will be defined as zero. Conversely, when the labels in $P_{frame}$ are all one, $P_{sp}$ is defined as 1. As the total number of frames is fixed, increasing $T_{s}$ will reduce the number of $D_{sp}$. This significantly affects the accuracy of the model, which will be confirmed in later experiments.

### 3.2. GCCM-Net

The 3D-CNN can extract spatiotemporal features in time continuous tasks, such as I3D [26] and X3D [25]. The input of a 3D convolution has one more dimension than 2D convolution. For video classification, this dimension is time. If the *j*th 3D feature map *v* in the *i*th network layer is $v_{ij}$, the 3×3×3 3D convolution can be mathematically expressed as
(1)$$\begin{array}{cc} v_{ij}^{xyz} & =\sum_{m}\sum_{p=1}^{3}\sum_{q=1}^{3}\sum_{r=1}^{3}\omega_{ij}^{pqr}v_{(i-1)m}^{\left(x+p-1)(y+q-1)(z+r-1\right)} \end{array}$$
where $\omega_{ij}$ is the kernel connected to the *m*th feature map $v_{(i-1)m}$, (x,y,z) is the value in the feature map, and (p,q,r) is the value in the kernel. As observed, computations increase with the expansion of the convolution kernel from 2D to 3D, which is inconvenient for deploying a real-time task related to robots. In this study, considering the continuous video processing, the GCCM-net was completely composed of 3D convolutions.

According to the above requirements, a compressed convolution layer (CCL [27]) is selected and improved, as shown in the lower left corner of Figure 6. Here, the convolution unit of CCL is upgraded from 2D to 3D. Furthermore, compared with two 3×3×3 convolution layers, CCL consists of a 1×1×1 convolution layer together with a 1×1×1 and 3×3×3 parallel convolution layers. Each parallel convolution of this layer accounts for only half the channels. The advantage of improving the CCL is that not only it reduces the volume of the model, but also improves the detection accuracy about 2%; the number of parameters declines by approximately 3/4 and the floating-point operations per second (FLOPs) diminishes by approximately 5/6, as Table 2 shows.

By observing the dataset characteristics in Figure 6 it can be concluded that the features of other positions in the image, except the deformation of the guidewire, are not required, and the pixel value of the unnecessary features is always zero. Therefore, this paper selects an improved spatial attention layer (SAL [28]) to strengthen the representation ability of the network, as shown in the lower right of Figure 6. In this study, the convolution unit of SAL is also upgraded from 2D to 3D. Furthermore, compared with 3×3×3 convolution layers, SAL adds a parallel layer of maxpooling and avgpooling. The function of this layer is to help the network focus on important features and suppress unnecessary ones [28]. The results from Table 2 show that SAL only slightly increased the parameters, but the increase in detection accuracy was approximately 1%.

The overall structure of GCCM-net is shown in the top half of Figure 6. It contains six layers, the first and the last layer are 3×3×3 convolution layers, and the middle four are improved layers with SAL and CCL. Each middle layer is downsampled by a Maxpooling layer. The final feature map is downsampled by an Avgpooling layer, in which all features are averaged into a binary result. For the training process, the datasets after fivefold cross-validation from $D_{sp}$ are regarded as training inputs. The cross-entropy loss and stochastic gradient descent (SGD) were also used to optimize the model. The number of training iterations was 20. Learning rate was 0.01, stride was 0.00001 and momentum was 0.9. The whole model was run under the PyTorch platform and Python framework, and the training time of each model was approximately 0.8 to 1 h on one RTX 3090 GPU.
micromachines-13-02237-t002_Table 2Table 2Ablation study results.ModelsParamsFLOPsFPSDsp1Dsp2Dsp3DtotalMobile-net [29]3.29 M11.1250.3490.58 ± 0.48%88.12 ± 0.81%86.21 ± 1.82%92.94 ± 0.41%Shuffle-net [30]0.95 M7.4753.7089.39 ± 0.47%87.55 ± 1.03%85.29 ± 1.30%92.26 ± 0.73%Squeeze-net [31]1.83 M37.55106.7888.10 ± 0.98%85.65 ± 1.53%81.84 ± 3.03%90.82 ± 0.44%CNN-net4.01 M181.07112.8289.55 ± 0.97%87.96 ± 0.44%85.59 ± 2.13%91.94 ± 0.90%SAL-net4.25 M187.8899.2791.10 ± 0.61%88.90 ± 0.42%86.36 ± 1.95%93.79 ± 0.25%CCL-net1.11 M30.7361.1991.50 ± 0.44%89.16 ± 1.07%88.53 ± 0.84%94.19 ± 0.91%GCCM-net1.24 M45.1389.7292.41 ± 0.31%89.68 ± 0.40%87.55 ± 1.71%94.86 ± 0.31%


### 3.3. Ablation Studies

Ablation studies selectively remove a component from the model and assess the ensuing change in results to prove the importance of that component. In this study, compared with GCCM-net, other models selectively replace only specific layers. For instance, the SAL-net replaces convolution layers with spatial attention layers, while CCL-net replaces convolution layers with compressed convolution layers. A CNN-net is also established, which only contains the obvious convolution layers. In addition, state-of-the-art models were compared, including Mobile-net [29], Shuffle-net [30] and Squeeze-net [31]. These models were chosen because they are widely used and specially optimized for the model volume. Note that the convolution layers in these state-of-the-art models were also upgraded from 2D to 3D. The ablation study results are shown in Table 2.

The percentage results in Table 2 indicate the binary classification accuracy of models under different datasets. First, in terms of number of parameters, CCL-net only has 1.11 M, which is approximately four times less than the 4.01 M of CNN-net. Moreover, the FLOPs of CCL-net is one-sixth smaller than that of CNN-net. In deep learning models, it can be seen that the data volume affects the model accuracy. The order of precision in each model is $D_{total}>D_{sp1}>D_{sp2}>D_{sp3}$. More importantly, compared with CNN-net, the accuracy in SAL-net increases by approximately 2% in every dataset. The GCCM-net, a model combining CCL and SAL achieves the best results, with 94.86% on $D_{total}$, with a parameter quantity of only 1.24 M. Compared with the general 3D-CNN, the GCCM-net has obvious advantages in parameter quantity and classification accuracy. Compared with the models of other studies, our proposed model also shows a cutting-edge performance, and a classification accuracy increase of approximately 2%.

Through CUDA acceleration, the average inference time of a well-trained GCCM-net model is about 89.72 FPS. Although CNN-net can reach 112.82 FPS, its accuracy is about 3% lower than GCCM-net. Compared with the state-of-the-arts, the GCCM-net proposed in this study has significant advantages in speed and accuracy. However, considering the image preprocessing methods and the loss of recognition accuracy, the whole GCCM is controlled to be stable at 20 Hz. Since the haptic feedback device used in this study has a high-sensitivity sensor (haptic resolution of approximately 1000 Hz), the feedback force is also down-sampled to 20 Hz and fed back to human hand in the follow-up experiments.

### 3.4. Class Activation Mapping

Some studies [32] have pointed out that the robustness of the model can be proved using class activation mapping (CAM). Therefore, we projected the convolution feature weights of the last layer in GCCM-net against the input samples, and then observed the activated areas of stacked images in different samples. In Figure 7, four types of results, including true positive (TP, $P_{true}=1,P_{pred}=1$), true negative (TN, $P_{true}=1,P_{pred}=0$), false negative (FN, $P_{true}=0,P_{pred}=0$) and false positive (FP, $P_{true}=0,P_{pred}=1$) are visualized, where $P_{true}$ represents the result of manual annotation, and $P_{pred}$ represents the result of GCCM-net prediction. The label of “1” indicates that the guidewire encounters the “hidden collision”, while the label “0” indicates that the guidewire is running smoothly. The TP samples show that, when collision occurs, the model focuses on the guidewire shadows, while for FN samples, the model focuses more on the guidewire tips. In TN samples, the stationary guidewire shadows may be not obvious, which lead to misjudgements. In FP samples, the collisions with the guidewire body also may lead to misjudges. However, compared with “hidden collisions”, this kind of collisions do not affect the guidewire operation. Concluding the above observations, CAM indicates that GCCM-net is sensitive to the deformation characteristics of moving guidewire. This implies that the proposed GCCM-net has the ability to recognize “hidden collisions” only by the abnormal deformations of guidewire. If this information can be observed in real time, then a guidewire compliant control method can be implemented.

## 4. Guidewire Compliance Control Strategy

There are various medical evaluation standards for endovascular surgery, such as global rating scales (GRS) [33]. However, the real-time surgical operation risk is a complex information. It is obvious that excessive operating force can easily cause vascular injury and even punctures. Jin et al. [8] suggest that surgery haptic forces always come from three aspects, namely the viscous force in blood flow, the friction force of surgical tools and the contact force on the vascular wall. The first two types of forces are unavoidable, but the contact force can be reduced by skillful procedures. For instance, Shi et al. [6] proposed a method of “Z-score”. If the “Z-score” exceeds a certain threshold, the robot determines that a collision has occurred, the feedback force value is doubled in time, and the operation risk will be effectively reduced. Bao et al. [7] indicated that the operation state can be divided into safe operation, potentially unsafe operation, and dangerous operation. In dangerous operations, the robot will generate resistance through its claws to prevent the guidewire from advancing. The above methods of surgical risk classification all depend on the prior knowledge of expert surgeons. This implies that an intraoperative condition needs to be given artificially. For example, the feedback force should not exceed 2 N. In contrast, the proposed GCCM-net is trained before surgery.

In this section, a surgical risk classification model is established based on the “hidden collisions” detected by GCCM-net. Next, a real-time, end-to-end guidewire compliance control strategy (GCCM-strategy) is established. The word “real-time” implies that GCCM-strategy is proposed as a strategy-based force feedback to the surgeon’s hand before the dangerous force occurs, which improves the efficiency and safety of the operation. The word “end-to-end” implies that the GCCM-strategy only accepts the information from the GCCM-net and robot, without adding artificial conditions during the operation.

### 4.1. Surgical Risk Classification

In most endovascular operations, the guidewire travel can proceed safely along straight pathways, but may become difficult in the branches. Therefore, the “hidden collisions” are an important watershed for dangerous and safe situations. To classify surgical risks in real time, a sketch map is drawn in Figure 8. In this figure, the operating risk is naturally divided into four conditions:Safe area A: The slave end drives the guidewire forward in the non-collision area. At this time, the load cell only measures the friction force.Risk area B: As the slave end continues to move, if the guidewire tip touches the vascular wall, the “hidden collisions” occurs. The feedback force becomes the sum of collision and friction forces.Dangerous area C: For pushing the guidewire forward through the vascular vessel, the operator rotates the slave end. At this time, the operation will be different according to the vascular morphologies. However, feedback force demonstrates a fluctuating state. In this case, the improper operation may cause danger, and the vascular wall may be punctured.Safe area D: The operator is aware of the danger and proceeds to retreat and adjust the movement.

Figure 9 shows a ten-second operation sequence in $D_{frame}$. The $F_{max}$ represents the maximum feedback force and the $S_{max}$ represents the maximum displacement of the slave end. According to Figure 8, the sequence is divided into four parts: Area A, Area B, Area C, and Area D. The safe operation is set to green, the risk operation is set to orange, and the dangerous operation is set to red. Based on force $F_{max}$, displacement $S_{max}$ and “hidden collisions”, the operation state of the guidewire can be clearly distinguished in real time. In this figure, the “hidden collisions” time is regarded as a risk switch, and the $F_{max}$ and $S_{max}$ are used to judge the degree of risk. Note that the source of $F_{max}$ not only includes the surgeon’s operation, but also may come from environmental factors in real surgery, such as vascular deformation, patient’s body movement, which will lead to the $F_{max}$ not always preceding $S_{max}$. If $F_{max}$ occurs before $S_{max}$, we are in Area C. Contrarily, if Area C and Area B coincide, risk and dangerous areas coexist. Because the vascular environment is complex and unknown, the safe control strategy designed by the force feedback sensor cannot cope with these changing operating situations. For instance, a sharp increase in force, or a sudden change in vessel branching, will both lead to the instability of the control of robot. So we do need to make a control strategy which combines the force and visual information.

### 4.2. GCCM-Strategy

Experienced surgeons avoid a risky operation to reduce the occurrence of high-risk forces. However, owing to the long operation time, physical fatigue might affect the surgeon’s judgment; therefore, it is necessary to establish a real-time strategy that always ensures the operation safety. With this in mind, we designed the GCCM-strategy. Schematic illustrations of the force feedback on the GCCM-strategy are shown in Figure 10.

Compared with other methods [6,7], the proposed GCCM-strategy includes a visual switch function, which is provided by the robot autonomy and without setting a threshold in advance. Therefore, in GCCM-strategy, the resistance Fe is only initialized at zero and updated during the real-time operation. What really interacts with the operator is K(Fe), where *K* refers to Kalman filter and Fe refers to the common compliance control force. The Kalman filter [34] can estimate the current system state by adding the weights of observation and prediction. In this study, it is used to reduce the effects of recognition noise, hand tremor, and mechanical vibration. When the collision occurs, robots with Fe prefer maintaining the guidewire tip stationary. The resultant force on the object can be mathematically expressed as
(2)$$\begin{array}{c} M\times S^{\prime \prime }+B\times S^{\prime }+A\times S=Fe \end{array}$$
where *M* represents the inertia coefficient, *B* represents damping coefficient, and *A* represents stiffness coefficient. Because the guidewire tip is soft, M=A=0. Then strategy based force Fe is only calculated by the operating speed $S^{\prime }$ and damping coefficient *B*. The damping coefficient *B* represents the energy dissipated by the vibration of objects. Here, it can be understood as a constraint condition for operating stability. If the real-time prediction result $P_{pred}$ from GCCM-net is regarded as guidewire tip’s damping coefficient, then Equation (Equation 2) is also expressed as
(3)$$\begin{array}{c} K\left(P_{pred}\right)\times K\left(S^{\prime }\right)=Fe \end{array}$$

This equation is the key to the GCCM strategy. If $P_{pred}=0$, regardless of what is the value of $S^{\prime }$, then Fe=0. In other words, in the case of safety, the operation is not affected by the GCCM-net, but when $P_{pred}=1$, the Fe shows a characteristic of viscous force. The faster the robot moves, the greater the force. If the operation stops, Fe=0; the robot cannot move autonomously. The GCCM-strategy establishes the relation between vision and force.

### 4.3. Evaluation Experiments

Based on ESR experimental platform mentioned in Section 2, a special operation task was designed and shown in the left half of Figure 11; it provides a small gap for the guidewire to pass through, simulating the carotid artery stenosis operation in endovascular surgery. This task is complex and dangerous for patients, and generally needs the assistance of angiography. With the GCCM-strategy, the operator can improve the efficiency of the operation in the invisible vascular environment. Therefore, it is more helpful to reduce the frequency of angiography in surgical practice and reduce the risk of surgery.

Five guidewire operators were invited to accomplish this experimental task. These operators were all novices and were from the non-medical field, and they had never operated the robot or other related equipment before. Two operating environments were proposed for this task: “Visible vascular” and “Invisible vascular”. Each environment had four operating conditions:Human hand (HH): Guidewire is directly controlled by human hands.Robot only (RO): Guidewire is controlled by robotRobot with force sensors (RF): The robot not only controls the guidewire, but also provides force feedback based on force sensors.Robot with GCCM-strategy (RG): The robot not only controls the guidewire, but also provides force feedback based on GCCM-strategy.

Therefore, there were eight tasks to be completed by each operator. Except for the vascular contour, the strategy used was not revealed to the operator, and a supervisor was appointed to oversee the whole process who also controlled the keyboard. To avoid unpracticed situations, each operator was allowed to operate the robot for 10 min before the task and was required to try three successful approaches. Finally, the operation trajectory with the shortest time in each case was filled in the results.

The robot operation results from one operator are shown in the right half of Figure 11. The small blue circles represent the positions of the guidewire tip. As expected, the circles in HH and RO cases were quite different. RF showed that the guidewire stays at obstacles for excessive time. This proved that force feedback reduced the operation speed and improved the stability. On the contrary, the case of RG showed that the control of guidewire was more submissive; it was easier to control the guidewire in the right direction, and smoothly pass it through obstacles without increasing force. These results proved that the GCCM-strategy could improve the efficiency and safety of the guidewire operation.

To further explain the principle of the GCCM-strategy, the real-time force changes of in RF and RG cases are exhibited in Figure 12. Because Fe is always greater than zero, to demonstrate the results more clearly, the *F* values also greater than zero are shown in the same figure. The force results show that in RF, the “hidden collisions” occurred many times, while in RG, it only occurred twice. This is consistent with Figure 11c,d. Excluding the possible luck of robot operators, Figure 12 shows obvious differences between strategy force and sensing force. Because GCCM-strategy was just observation and did not take place in RF. It was a high delay between the peak values of strategy force and sensing force. However, in RG, the changes of both forces tended to be consistent. This result was confirmed many times, indicating that GCCM-strategy could prevent the possible risks in guidewire operation. In addition, the Kalman filter reduced the force vibration. If the force coincidence is strictly considered, the false positive and true negative samples of GCCM-strategy may also affect the operation. In Figure 12a, there were several false strategy force peaks, but this did not cause the fluctuation of sensing force. This means that false positive samples may hinder the robot’s operation, but there is no evidence that false positive samples increase the surgical risk. As for true negative samples, there were only two in Figure 12a,b and the sensing forces did not exceed 300 mN; hence, the impact on operation safety was also within an acceptable range.

Calculating the rotation time of the guidewire can also reflect the efficiency of the method. Therefore, a boxplot for the rotation time from five operators is drawn in Figure 13. HH is missing because the rotation could not be recorded. Among the remaining three cases, RG achieved the shortest average rotation operation time (6.52 s in invisible and 4.86 s in visible vascular operation). Compared with RF (8.56 s in invisible and 6.90 s in visible vascular operation), there was an improvement of approximately 23% to 29% using the GCCM method. These results show that the GCCM-strategy could improve the efficiency of the guidewire operation.

Table 3 shows all indicators calculated for the five operators. Three of them can be measured directly, such as average operation time *t* (starts when the speed is above zero and ends when the guidewire tip reaches the target), average *F* (from force sensors), and average operation speed $S^{\prime }$ (from the slave robot). The other two indexes, average $P_{pred}$ observed by GCCM-net and average Fe obtained from GCCM-strategy, must be estimated. Note that when human hands directly operate the guidewire, *F* and $S^{\prime }$ cannot be measured. In addition, there was no *F* feedback on hands in feedback-free cases, and only the relevant values were counted.

In general, HH had the best flexibility and could complete the tasks in the shortest time. According to Equations (2) and (3), the stability of the operation can also be measured by the predicted value of $P_{pred}$. If the $P_{pred}$ value is too large, it proves that the probability of “hidden collision” is higher, and the contact between the guidewire tip and the blood vessel wall is more frequent during the operation. This is not in line with our expected goal of “guidewire passing through vascular branches smoothly”. In Table 3, when the vascular are visible, the results of $P_{pred}$ values are similar in four cases. This proves that the operation stability can be greatly improved by using visual information. The behavior of the guidewire passing through the vascular branch is no longer “the exploration behavior in unknown space”. However, when the vascular is invisible (as mentioned before, this is a common situation in X-ray surgery), the RG’s $P_{pred}$ values is the smallest. This means that the results of deep learning model recognition has a positive impact on the operation behavior.

It is generally considered that it is meaningful to improve the safety of operation by using a shorter operation time and outputting less operation force. Compared with RF, RG reduced 44.0% of the average contact force while shortening 24.6% of the average operation time, which confirmed the effectiveness of proposed method. In summary, for the five operators involved in the experiment, GCCM improved their ability to control the guidewire and reduced the operation risks.

## 5. Discussion on the Application of GCCM

The guidewire compliant control method (GCCM) proposed in this paper consists of two parts: GCCM-net and GCCM-strategy. The goal of GCCM is to avoid “hidden collisions” in an invisible vascular environment, thus improving the efficiency and safety of guidewire operations. In the proposed ESR experimental platform, our method achieved the expected effect. However, considering the complexity of real surgery, the applications of GCCM are worth further study.

The proposed ESR experimental platform has the functions of haptic force perception and real-time monitoring. However, there is still a gap between the vascular phantom image and the fluorescence image. Owing to the limitation of experimental conditions, it is currently difficult to obtain a large number of robot-assisted surgical images. In addition, the model proposed in this paper may work on a non-fluorescent environment, which means that the training process can be performed through the guidewire mask only. Therefore, the real image is weakened in importance.

As for the proposed GCCM-net, this paper optimized the model accuracy and real-time performance through specific deep learning structures, but the errors deserved more attention. Although the deep learning model can do well in some tasks, it still lacks interpretability. The purpose of defining a classification problem of “hidden collisions” is that it can be regarded as a high-dimensional feature with expert experience. In the future, the behavior of multi-instrument interactions, especially in invisible vascular environment, will be worthy of observation and recognition by robots.

The proposed GCCM-strategy proves the importance of high-dimensional feature learning in ESR systems and provides a solution for upgrading the robot function. Compared with the load cells, the strategy-based force feedback can prevent the operation risks in advance, which is crucial for operation safety. Two more drawbacks are worth mentioning. First, the tactile device proposed in this paper is not a real guidewire, which entails a lower operating efficiency than human hands. Second, none of the operators invited for this study had previous experience in surgery. It is certain that regardless of surgical experience, the robot-assisted methods can avoid physical fatigue caused by concentration for a long time. For experienced surgeons, the effect of the active intervention method needs further study.

## 6. Conclusions

In this paper, a deep learning-based compliance control method (GCCM) was proposed to improve the efficiency of guidewire operation, which included the GCCM-net and GCCM-strategy. Compared with the baseline and the state-of-the-art methods, the GCCM-net had a better ability to observe whether the guidewire collides with the vascular wall and to classify the operation risk in real time. The vascular phantom experiments showed that the accuracy of the proposed deep learning model could reach 94.86 ± 0.31%. At the same time, with GCCM-strategy, the guidewire had higher maneuverability. The average operating force was reduced by 44.0% and the average operating time was reduced by 24.6%. It was proved that the combination of visual information with force information had a positive influence on endovascular surgery robot performance. Future research should focus on real fluorescence images and evaluation of the proposed method by experienced surgeons.

## Figures and Tables

**Figure 1 micromachines-13-02237-f001:**
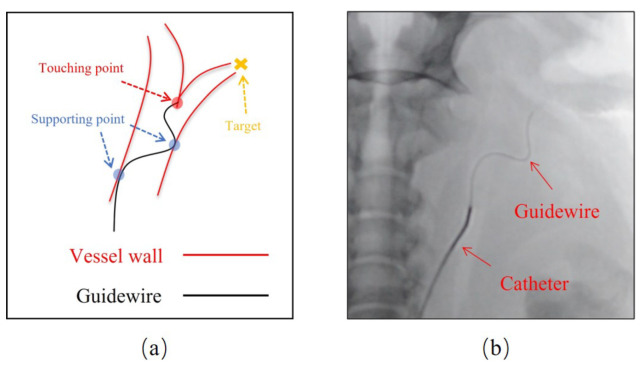
Surgeons operate guidewires to explore the access with limited surgical vision. (**a**) Hidden collision. (**b**) Fluorescence imaging.

**Figure 2 micromachines-13-02237-f002:**
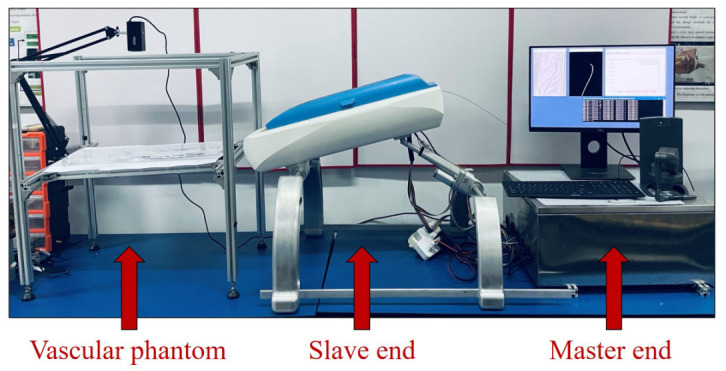
The robot experimental platform.

**Figure 3 micromachines-13-02237-f003:**
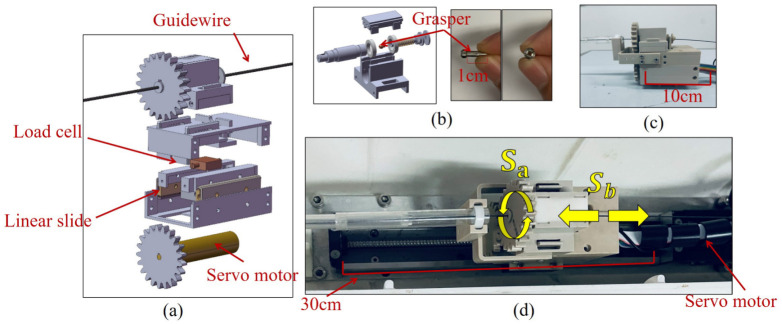
The slave end drives the guidewire to translate and rotate through a tiny gripper. (**a**) Structure diagram. (**b**) Guidewire gripper. (**c**) Prototype (**d**) Slave end.

**Figure 4 micromachines-13-02237-f004:**
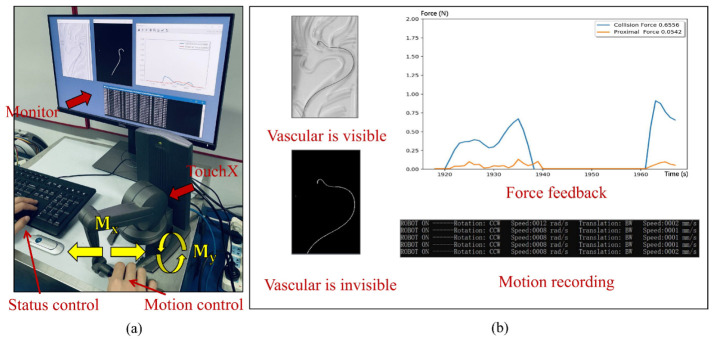
The master controller and force feedback. (**a**) Master end. (**b**) Real-time surgical state.

**Figure 5 micromachines-13-02237-f005:**
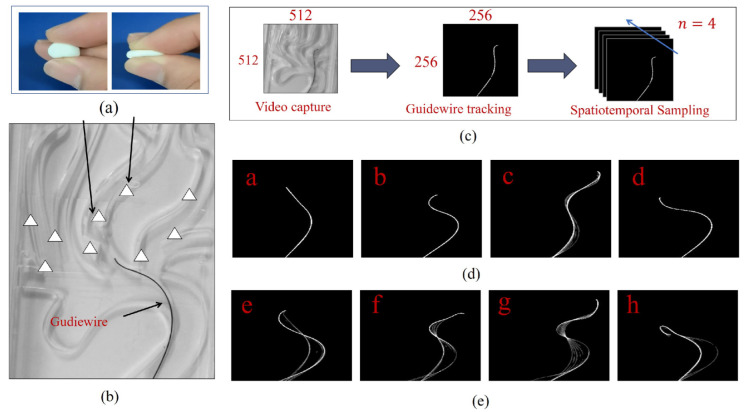
The vascular phantom and “hidden collisions” samples. (**a**) Soft white clay. (**b**) Vascular phantom. (**c**) Image processing and sampling. (**d**) Non-collision samples. (**e**) “Hidden collision” samples.

**Figure 6 micromachines-13-02237-f006:**
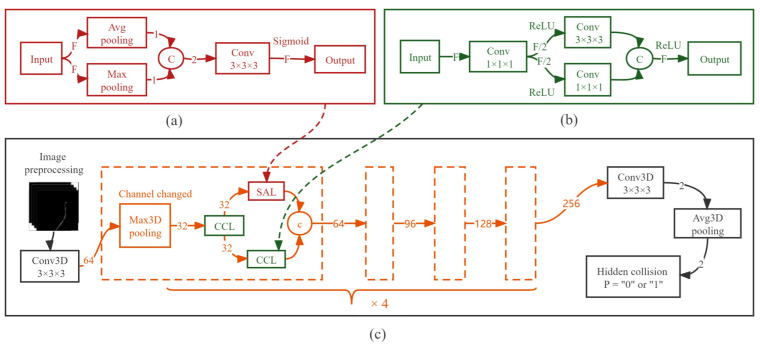
The GCCM-net model accepts the preprocessed frames of the guidewire outline and outputs the probability value of the hidden collision. A complete GCCM-net contains four repeated and serial modules. Each module contains two CCLs and one SAL. There is a 3D Maxpooling layer in front of the module to adjust the number of channels F. The input data have a size of (B, F, x, y, z), where B represents the number of batches, F represents the number of feature channels, and x, y, z are from four consecutive frames of images. (**a**) Spatial attention layer (SAL). (**b**) Compression convolution layer (CCL). (**c**) Deep learning network in guidewire compliant control method (GCCM-net).

**Figure 7 micromachines-13-02237-f007:**
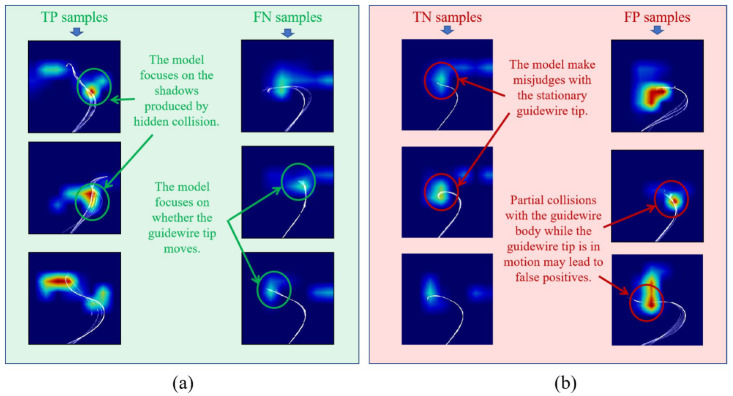
Class activation maps. We visualize the features of the last layer of GCCM-net and superimpose them on the original image (3D images are also superimposed). The red color in the figure represents the area where the weight is activated, and the blue color represents the area that is not activated. Where (**a**) represents correctly identified TP and FN samples, and (**b**) represents incorrectly identified TN and FP samples.

**Figure 8 micromachines-13-02237-f008:**
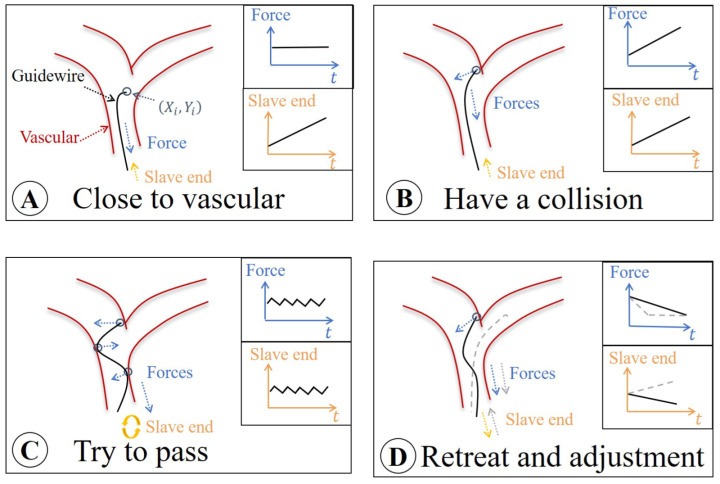
Sketch map of guidewire operation in vascular branch.

**Figure 9 micromachines-13-02237-f009:**
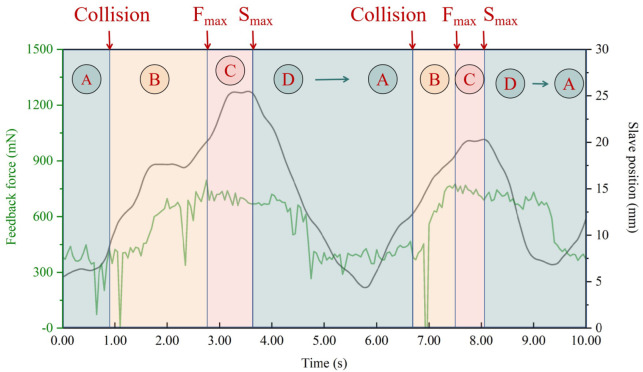
The Real time surgical state classification.

**Figure 10 micromachines-13-02237-f010:**
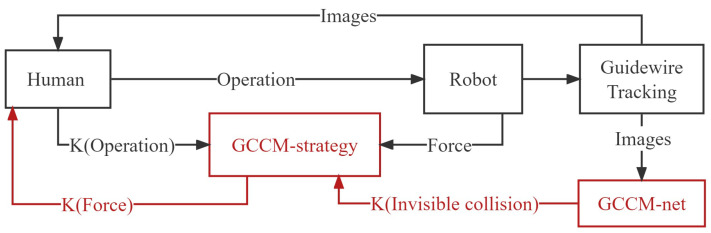
Flow charts of sensors-based and strategy-based force feedback.

**Figure 11 micromachines-13-02237-f011:**
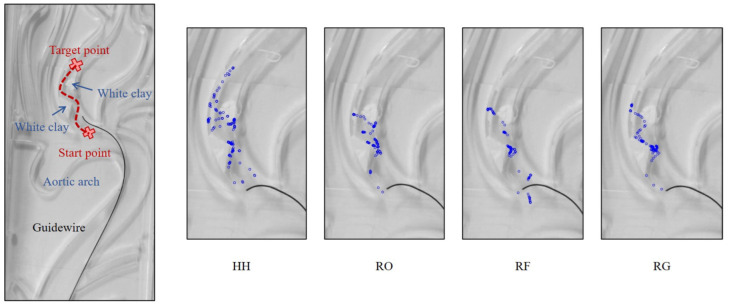
Comparison of surgical trajectories.

**Figure 12 micromachines-13-02237-f012:**
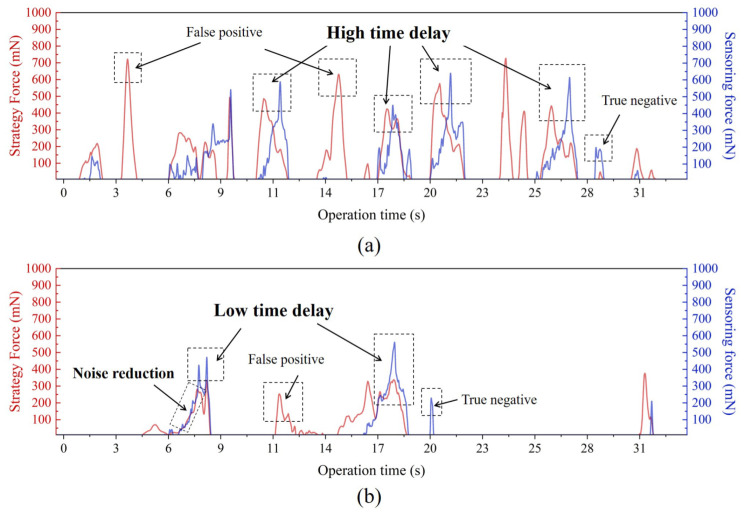
Comparison of feedback force. (**a**) Without GCCM in RF. (**b**) With GCCM in RG.

**Figure 13 micromachines-13-02237-f013:**
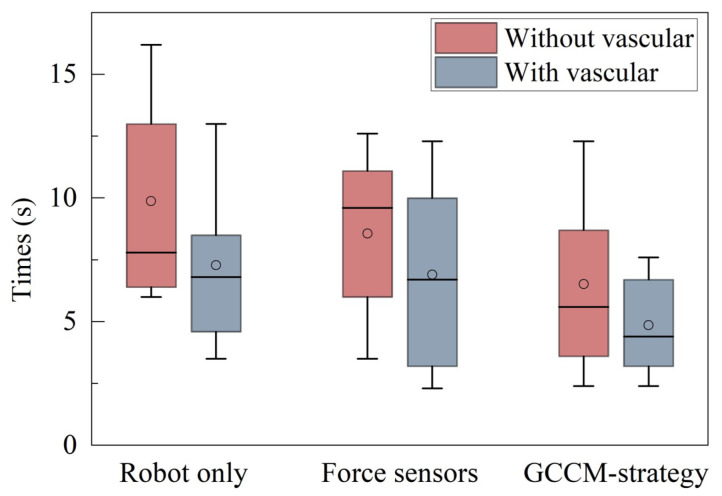
Rotation time from five operators.

**Table 1 micromachines-13-02237-t001:** The dataset statistics.

Name of Datasets	$T_{f}$	$T_{s}$	$T_{d}$	Number of Samples	
				$P_{\mathrm{sp}}=0$	$P_{\mathrm{sp}}=1$
$D_{sp1}$	1f	4f	4f	8578	5664
$D_{sp2}$	2f	8f	8f	4085	2736
$D_{sp3}$	4f	16f	16f	1867	1230
$D_{total}$	-	-	-	14,530	9630

**Table 3 micromachines-13-02237-t003:** The result of system evaluation.

Environments	Conditions	*t* (s)	*F* (mN)	$S^{\prime }$ (mm/s)	$F_{e}$ (mN)	$P_{\mathrm{pred}}$ (%)
Visible	HH	10.6	-	-	-	41.1
	RO	33.0	137.7	27.1	445.0	41.2
	RF	35.5	113.1	28.9	301.7	31.6
	RG	28.6	84.5	20.1	248.3	36.0
Invisible	HH	17.4	-	-	-	50.6
	RO	36.8	275.1	26.2	498.8	58.5
	RF	31.2	240.9	27.5	567.4	51.2
	RG	23.5	134.7	23.4	216.9	36.3

## Data Availability

The authors are unable or have chosen not to specify which data has been used.

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
