# Peer review of "A Deep-Learning-Based Guidewire Compliant Control Method for the Endovascular Surgery Robot"

_micromachines, 2022, doi:10.3390/mi13122237_

Round 1
Reviewer 1 Report
Summary:
This paper proposes a deep learning-based guidewire compliant control method (GCCM) for endovascular surgery robot (ESR) systems. The main contributions of this paper are (1) a new deep learning model named GCCM-net that is effective in real-time detection of any collision between guidewire tip and vascular wall, (2) a real-time operational risk classification method named GCCM-strategy is proposed. When surgical risks occur, the GCCM strategy uses the result of GCCM-net as damping and decreases the robot's running speed through virtual resistance, (3) the proposed method significantly reduces the average operating force and operating time.
Strength:
The proposed approach of the combination of visual information with force information shows promising performance in achieving higher guidewire maneuverability, especially in terms of operating force and time reduction. Comprehensive ablative studies and class activation mapping were also used to evaluate the proposed deep learning network and the approach of labels assigned. The evaluation of different systems validates the effectiveness of GCCM-strategy in improving the efficiency and safety of the guidewire operation.
Weakness:
(1) As the methods are constrained in the invisible vascular environment, there could have been more said over the advantage and disadvantages of operation in both invisible and visible vascular environments. Does the operation this way provide more convenience than guidewire segmentation in a visible vascular environment?
(2) A lack of a certain threshold of abnormal bending in "hidden collision" might limit the path selection of guidewire.
(3) GCCM-net illustration is not clear enough to identify which parts correspond to CCL and Sal modules. And the network inference time should be included such that the real-time could better be justified.
(4) Some errors occur in the Class Activation Mapping section where the definition of types of results is confusing.
Reviewer 2 Report
Thank you for giving me the opportunity to review this interesting paper featuring deep learning method to detect obstacles (vessel branching) during the endovascular surgery. This is indeed an important topic, and with potentially high benefits to patients. Few minor issues:
-Line 105. M(x,y)T was never properly defined.
-Readers would greatly benefit if you can better explain difference between Dframe and Pframe. Pframe can only have a binary values. What exactly Dframe describes? Lines 148-152.
-Line 248. Could you please define Ptrue and Ppred?
-Line 62-64. Could you please re-phrase the sentence? It is not very clear. “If only observe the guidewire mask in invisible vascular environment…”
-Lines 313-314 (the last sentence of that paragraph). Could you please double check the grammar? The sentence is not clear.
Lines 410-412. What is meant by the “best stability”? Could you please explain as it is not clear, i.e. other parameters are measured, and the measure “stability” is introduced with conclusions.
It would have been indeed very interesting if indeed trained surgeon was brought to perform the tasks, and compared his performance to the GCCM-strategy. From the results and table 3 can be concluded that GCCM-strategy is only ~8% better than the non-trained person.
Round 2
Reviewer 1 Report
There is no other comments in this turn.